# Ultrafast carrier thermalization in lead iodide perovskite probed with two-dimensional electronic spectroscopy

Johannes M. Richter[1], Federico Branchi[2], Franco Valduga de Almeida Camargo[2], Baodan Zhao[1], Richard H. Friend [1], Giulio Cerullo[2] & Felix Deschler [1]

In band-like semiconductors, charge carriers form a thermal energy distribution rapidly after optical excitation. In hybrid perovskites, the cooling of such thermal carrier distributions occurs on timescales of about 300 fs via carrier-phonon scattering. However, the initial build-up of the thermal distribution proved difficult to resolve with pump–probe techniques due to the requirement of high resolution, both in time and pump energy. Here, we use two-dimensional electronic spectroscopy with sub-10 fs resolution to directly observe the carrier interactions that lead to a thermal carrier distribution. We find that thermalization occurs dominantly via carrier-carrier scattering under the investigated fluences and report the dependence of carrier scattering rates on excess energy and carrier density. We extract characteristic carrier thermalization times from below 10 to 85 fs. These values allow for mobilities of 500 cm$^2$ V$^{-1}$ s$^{-1}$ at carrier densities lower than $2 \times 10^{19}$ cm$^{-3}$ and limit the time for carrier extraction in hot carrier solar cells.

[1] Cavendish Laboratory, University of Cambridge, JJ Thomson Avenue, Cambridge CB3 0HE, UK. [2] IFN-CNR, Dipartimento di Fisica, Politecnico di Milano, Piazza L. da Vinci 32, 20133 Milano, Italy. Johannes M. Richter and Federico Branchi contributed equally to this work. Correspondence and requests for materials should be addressed to G.C. (email: giulio.cerullo@polimi.it) or to F.D. (email: fd297@cam.ac.uk)

Hybrid perovskite semiconductors have attracted strong scientific interest for optoelectronic applications. Simple and low-cost solution-based fabrication techniques can be used to obtain poly-crystalline thin films[1–3], as well as single crystals[4] and colloidal nanostructures[5]. Reports on sharp absorption onsets, long carrier lifetimes and high photo-luminescence quantum efficiencies[4, 6–11] highlight the semiconducting quality of these materials. Despite the current drive for higher optoelectronic device efficiencies and stabilities, little information exists about the fundamental non-equilibrium interactions of photo-excited charge carriers in these semiconductors, which define, for example, the fundamental limits of charge transport.

Photo-excitation of a semiconductor with free band continuum states leads to an electron population in the conduction band. Initially, the excitations are in a superposition of ground and excited states. Over the dephasing time these coherences are lost and a pure population is left. However, after light absorption this carrier population is not in thermodynamic equilibrium immediately, as the energy of the excited carriers matches that of the absorbed light. The spectrum of the excitation pulse and the selection rules of the semiconductor determine the initial energetic distribution of the carrier population, which can lead to the observation of a "spectral hole" in the absorption spectrum for narrowband excitation[12, 13]. Ultrafast scattering processes, such as carrier-carrier or carrier-optical-phonon scattering, lead to a broadening of the energetic distribution of the carrier population[14–16], which can eventually be described by an equilibrium carrier temperature higher than the lattice temperature. This process of exchange of energy among charge carriers is called thermalization. Subsequently, a cooling of the carriers occurs through carrier-phonon and carrier-impurity scattering processes, bringing the carriers and lattice to thermodynamic equilibrium[14]. The carrier thermalization and cooling processes after light absorption depend on the properties of the excited charge carriers and the band structure of the semiconductor. In optoelectronic applications, the carrier scattering rates determine the fundamental limits of carrier transport and electronic coherence.

For the prototypical semiconductor GaAs, studies of thermalization dynamics reported timescales ranging from 100 fs to 4 ps[12, 15, 17], and provided insights into dephasing times[12], band structure[15] and carrier-carrier scattering processes[18], which affect subsequent carrier cooling and recombination processes. In hybrid perovskites, ultrafast transient absorption spectroscopy has been used to study the carrier cooling process, which was found to occur on 100 s of femtoseconds timescales[19–22], with a strong contribution from a hot phonon effect at high fluences[23]. The thermalization process, on the other hand, has not yet been experimentally resolved and is expected to occur on much faster timescales[14–16].

To resolve ultrafast thermalization dynamics, the time evolution of the time-dependent energy distribution of an initial population excited at a well-defined energy needs to be tracked[24]. The necessary femtosecond time resolution requires the use of ultrashort, broadband light pulses, which compromises the desired high resolution in excitation energy. Two-dimensional electronic spectroscopy (2DES) is an extension of traditional transient absorption techniques that achieves both conditions simultaneously[25] by using a Fourier transform (FT) approach. In 2DES, two pump pulses are used instead of one, with the time delay between them (labeled $t_1$) being scanned with interferometric precision for a fixed delay between the second pump and the probe (the waiting time, labeled $t_2$). Taking the FT of the nonlinear optical signal over $t_1$ provides the excitation frequency axis, with a resolution that is only limited by the $t_1$ scanning range.

Here, we report 2DES experiments on lead-iodide hybrid perovskites using thin films of the prototypical material $CH_3NH_3PbI_3$. We extract thermalization time constants in the range of below 10–85 fs, depending on carrier density and excess energy and find that the main thermalization process is carrier-carrier scattering.

## Results

**Pump–probe experiment with sub-10 fs pulses.** We prepare thin films of $CH_3NH_3PbI_3$ on 170 μm thick glass substrates. Figure 1a shows the absorption spectrum of the samples with a bandgap around 760 nm (1.63 eV), which contains contributions from an excitonic transition near the bandgap and free carrier absorption towards shorter wavelengths[26].

We perform pump–probe experiments using sub-10 fs laser pulses, with a spectrum spanning from 550 nm (2.25 eV) to 750 nm (1.65 eV), as shown in Fig. 1a. All measurements were performed in the linear excitation regime as can be seen in Supplementary Fig. 2a and in conditions under which the samples showed a high photo-stability (see Supplementary Fig. 2b). Figure 1b shows the differential transmission ($\Delta T/T$) spectrum as a function of pump–probe delay and probe wavelength. For positive time delays the signal is dominated by a strong photo-bleaching (PB, $\Delta T/T > 0$) between 600 and 750 nm, which we attribute to a phase space filling of the electronic states and thus reduced absorption near the band edge due to excitation of

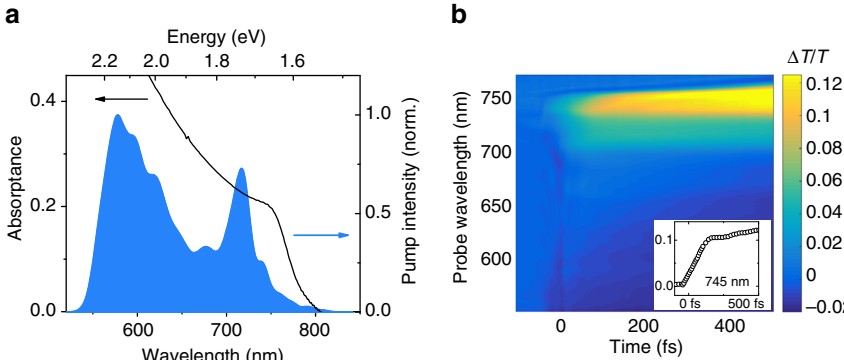

**Fig. 1** Pump–probe spectroscopy of lead iodide perovskite. **a** Absorption spectrum of lead iodide perovskite and broadband laser spectrum used for the degenerate pump–probe and 2D electronic spectroscopy (2DES) experiments. The pump spectrum overlaps with the free band continuum as well as the excitonic transition of perovskite. **b** Pump–probe spectroscopy of perovskite: $\Delta T/T$ spectrum as a function of probe wavelength and pump–probe delay (excitation density: $2 \times 10^{18}$ cm$^{-3}$). The broadband nature of the short pump pulses makes the observation of a non-thermalized distribution and thermalization difficult in pump–probe experiments. Inset: Pump–probe dynamics at 745 nm probe wavelength. The signal rises over two distinct timescales

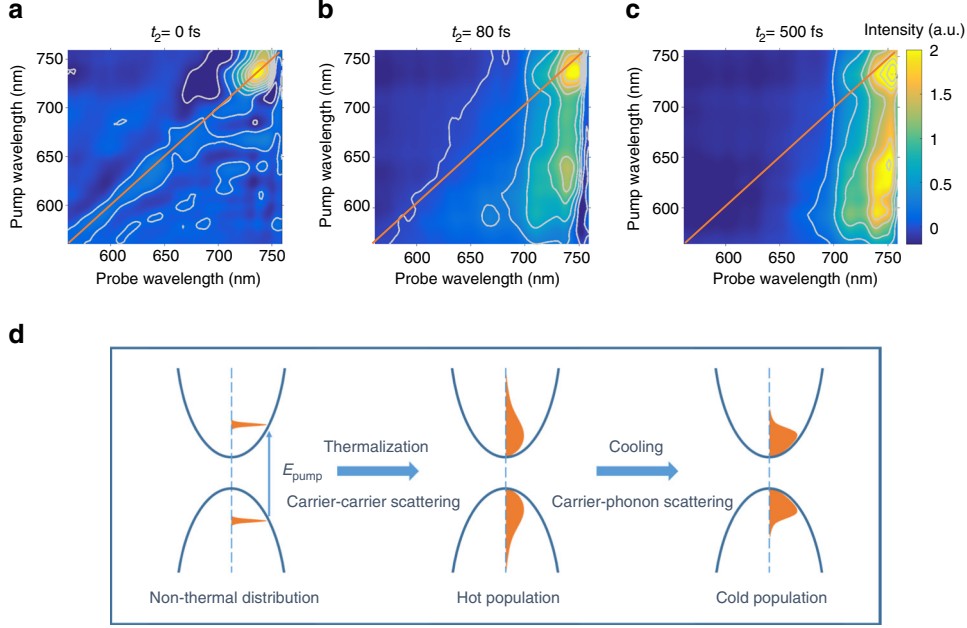

**Fig. 2** Carrier relaxation probed with 2D electronic spectroscopy. **a–c** 2DES maps for time delays $t_2$ of **a** 0 fs, **b** 80 fs and **c** 500 fs for an excitation density of $2 \times 10^{18}$ cm$^{-3}$. The diagonal of the 2DES maps is indicated with orange lines. **d** Schematic illustration of carrier relaxation processes. Initially, a non-thermal carrier energy distribution is excited. After undergoing carrier-carrier scattering, carriers form a thermalized distribution with a temperature higher than the lattice. Through carrier-phonon scattering, the carriers subsequently cool down until they reach an equilibrium with the lattice temperature

carriers in the band-to-band continuum. The broad negative signal between 550 and 650 nm overlapping the PB has been shown to originate from a transient change in reflectivity[18, 19]. At negative time delays, i.e. when the probe pulse arrives before the pump, we observe spectral oscillations (see Supplementary Fig. 3 for detailed map) which display an increasing period when approaching time zero[14, 27, 28], defined as the point of temporal overlap of pump and probe pulse. These oscillations, known as the pump-perturbed free-induction decay, originate from a transient grating signal between pump and probe, which is emitted along the probe's propagation direction and is only present at negative time delays[13, 29, 30]. It will therefore not affect the analysis of the signal at positive time delays.

The inset in Fig. 1b shows the $\Delta T/T$ dynamics near the band edge at 745 nm probe wavelength. We see a rise in the PB signal with two distinct components: A fast rise within the first 130 fs and a slower rise longer than the 500 fs maximum time delay. While the second component is consistent with the reported carrier cooling times[19], the first component is likely to be due to carrier thermalization. This signal contains contributions from carriers excited at all energies within the broad pump pulse spectrum, which thermalize to a distribution peaked close to the bandgap. For this reason, pump–probe spectroscopy with broadband pulses is unable to resolve an excitation energy-dependent thermalization process. At the same time, broadband pump pulses are required in order to provide the time resolution necessary to measure the observed ultrafast thermalization process.

**Carrier relaxation probed by 2DES spectroscopy.** In order to achieve a high resolution in pump energy while maintaining ultrafast time resolution, we perform 2DES experiments with the same broadband sub-10 fs pulses employed for pump–probe. An illustration of the 2DES setup layout can be found in Supplementary Fig. 1 and a detailed description in Supplementary Note 2. We perform 2DES measurements in a range of the

waiting time $t_2$ (which corresponds to the pump–probe delay) from −100 to 500 fs and for excitation densities of $2 \times 10^{18}$ cm$^{-3}$ and $2 \times 10^{19}$ cm$^{-3}$. The full sets of 2DES maps are available as video files (Supplementary Movies 1 and 2). The lower fluence measurement is performed at an excitation density just below the onset of the hot phonon effect (re-absorption of excited phonons by charge carriers)[19], which allows us to observe carrier cooling. The higher fluence reaches an excitation density where the hot phonon effect slows down carrier cooling, preventing its observation within the 500 fs measurement window. In order to check the quality of the computed 2DES maps, we plot the projection of the 2DES data along the pump wavelength axis in Supplementary Fig. 4 and compare it with the pump–probe data. We find a very good agreement between the two measurements indicating a high reliability of the 2DES data.

At negative time delays, we observe spectral oscillations, which again can be identified by their increasing period when approaching time zero. These are likely to be due to the pump-perturbed free induction decay. However, the current scanning of $t_1$ in our 2DES setup is such that the pulse sequence is changed for negative $t_2$ time delays, which complicates the interpretation of the data in this regime beyond the scope of this report.

At a time delay $t_2 = 0$ fs, we observe a PB signal dominantly along the diagonal of the 2DES map as seen in Fig. 2a for a carrier density of $2 \times 10^{18}$ cm$^{-3}$. This signal originates from a non-thermalized carrier distribution, which has not yet undergone scattering events, so that we observe a PB at the same wavelengths at which we excite with the pump pulse.

Within the first 100 fs after excitation, we observe a spectral broadening of the PB signal for each pump wavelength as seen for $t_2 = 80$ fs in Fig. 2b, indicative of carrier thermalization, i.e., the exchange of energy amongst charge carriers. After completion of the thermalization process, the carrier population can be described by a carrier temperature $T_C$ but still remains out of thermodynamic equilibrium with the lattice (typically $T_C > T_L$ where $T_L$ denotes the lattice temperature). The carrier energy distribution and thus the $\Delta T/T$ signal at high probe energies

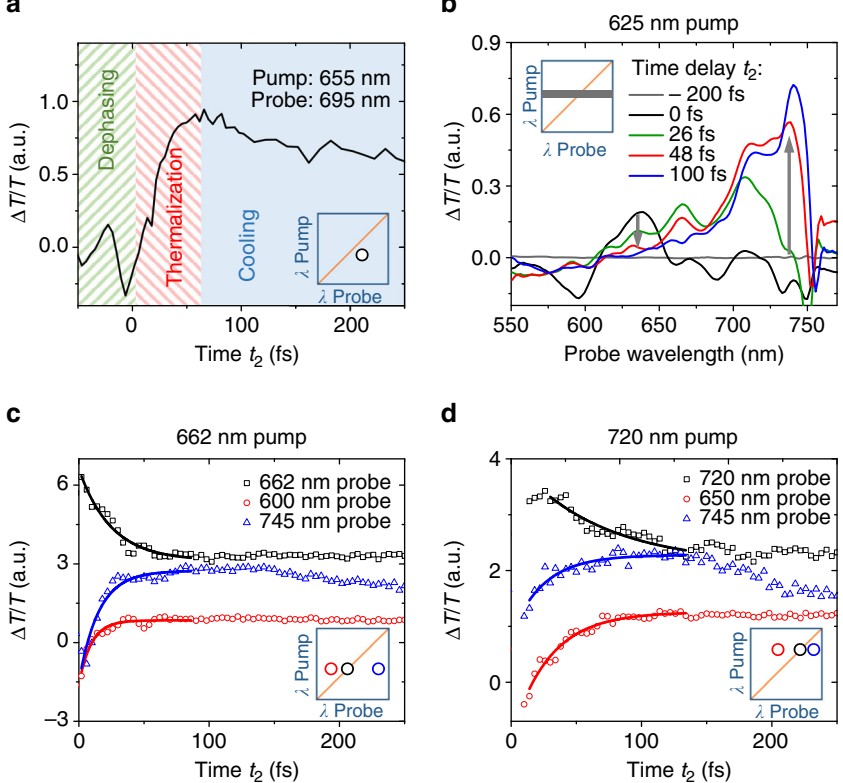

**Fig. 3** Carrier thermalization in perovskites. **a** 2D electronic spectroscopy (2DES) kinetic at 655 nm pump and 695 nm probe wavelength, which is on the right side of the 2DES map diagonal and thus between the pump wavelength and the bandgap (excitation density: $2 \times 10^{18}$ cm$^{-3}$). We observe three different regimes: a coherent regime at negative times during which we observe spectral oscillations, a thermalization regime during which we observe a rise in signal much slower than the temporal width of the instrument response function and delayed relative to the rise of the diagonal, and a carrier cooling regime during which the signal slowly decays. **b** $\Delta T/T$ spectra for 625 nm pump wavelength extracted from 2DES maps for different time delays $t_2$ at an excitation density of $2 \times 10^{18}$ cm$^{-3}$. Initially, we observe a peak around the pump energy. Carriers quickly thermalize and form a Boltzmann distribution. **c**, **d** Kinetics of carrier thermalization for **c** 662 nm and **d** 720 nm pump wavelength at an excitation density of $2 \times 10^{19}$ cm$^{-3}$. While the diagonal signal decays, the off-diagonal signals rise indicating that carriers scatter from the initial sharp energy distribution into a broad statistic energy distribution. The lines represent a mono-exponential fit to the experimental data

(energies larger than 1.7 eV or 730 nm) are expected to follow a Boltzmann function, according to[19]

$$\frac{\Delta T}{T}(E) \sim \exp\left(-\frac{E - E_f}{k_B T_C}\right) \quad (1)$$

where $k_B$ denotes the Boltzmann constant and $E_f$ the Fermi energy. We fit Eq. (1) to the $\Delta T/T$ spectrum at 625 nm pump wavelength and a delay of $t_2 = 52$ fs (Supplementary Fig. 5), from which we derive a carrier temperature of 1600 K. At later time delays, carriers undergo carrier-phonon scattering, which will bring the excited carrier population into an equilibrium with the lattice and cool the carrier temperature $T_C$ down to ambient levels (about 300 K). The cooling time has been reported to be around 200–400 fs below the onset of the hot phonon effect[19, 20]. At $t_2 = 500$ fs time delay, we extract a carrier temperature of 890 K (Supplementary Fig. 5) showing the progressive carrier cooling in good agreement to published carrier cooling times[19–21]. This confirms that the lower fluence measurement was performed below the onset of the hot phonon effect. At a time delay $t_2 = 500$ fs, we observe a carrier population which has now significantly cooled down compared to the carrier distribution at 80 fs, so that the 2DES map is now a vertical stripe corresponding to the band edge in Fig. 2c.

The different stages of carrier relaxation are illustrated in Fig. 2d. The clear separation in the time scales of the different relaxation processes can be seen in Fig. 3a by monitoring the

dynamics of the peak in the 2DES map corresponding to 655 nm pump and 695 nm probe wavelength: At negative time delays, we observe spectral oscillations characteristic of the pump-perturbed free induction decay. At positive time delays up to 100 fs, the PB signal rises due to thermalization while it subsequently decays when carriers cool to the band edge under carrier-phonon scattering. This is consistent with the observed two regimes in the rise of the bandgap signal in the inset of Fig. 1b.

**Investigating carrier thermalization.** Figure 3b shows the time evolution of the $\Delta T/T$ spectra at 625 nm pump wavelength, extracted from the 2DES measurements. Around 0 fs, we observe a peak in the spectrum near the pump wavelength of 625 nm. On either side of this peak, we observe a negative transmission change. For delay times close to the temporal overlap of pump and probe, this effect has been observed for GaAs before and was attributed to many-body edge singularities due to the non-equilibrium distribution function under photoexcitation[14]. This effect might also reduce the total photo-bleach intensity at time zero compared to later time delays. Other contributing factors will be a negative signal due to a change in reflectivity of the perovskite film[19], which can also be seen at 500 fs time delay as plotted in Fig. 2c in the probe wavelength range of 550–650 nm. At positive time delays, the peak in Fig. 3b at 625 nm probe wavelength decays, while the signal on either side of the peak rises. This shows that carriers are now occupying a broader

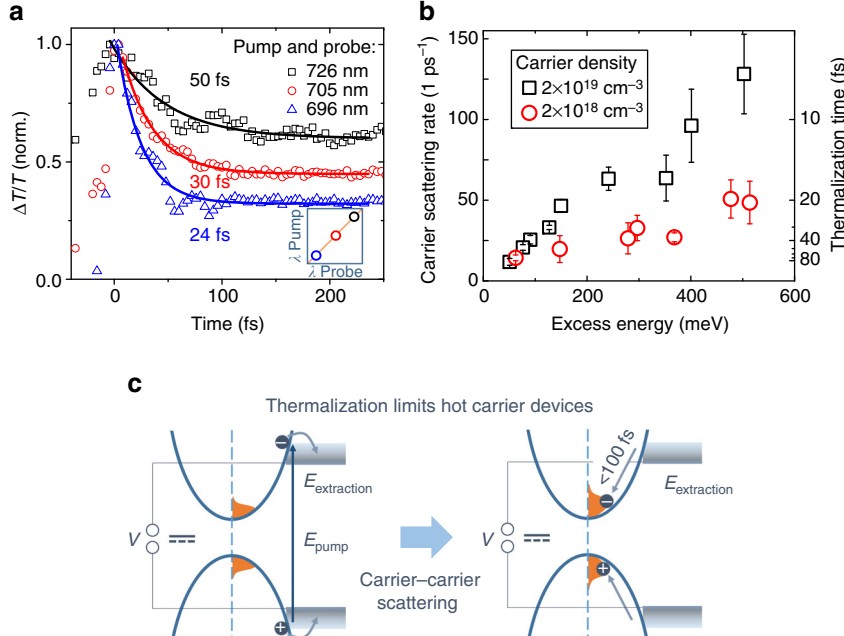

**Fig. 4** Pump wavelength dependence of carrier thermalization. **a** Pump wavelength dependence of the initial decay of the diagonal signal at an excitation density of $2 \times 10^{19}$ cm$^{-3}$. The shorter the pump wavelength, the faster the diagonal decays. The solid lines represent a mono-exponential fit to the experimental data. **b** Carrier scattering rate vs excess energy over the bandgap at 1.63 eV. The scattering rates increase with excess energy. We observe that the scattering rate shows a fluence dependence, which suggests that the main thermalization process is carrier-carrier scattering. The error bars represent the error of the mono-exponential decay time fits. **c** Schematic illustration of carrier thermalization under continuous wave illumination in a hot carrier extracting device. Due to the fast timescales of cooling compared to carrier recombination, there will always be a cold carrier population in a perovskite device. This population will quickly thermalize with any newly excited charge carriers. The thermalization time is therefore the limiting factor for hot carrier extracting devices

energetic range of states. We interpret this as carriers undergoing scattering processes, which will eventually lead to a thermal distribution. The peak of the spectrum is therefore now close to the band edge at 760 nm.

Carrier thermalization can be visualized from the dynamics in Fig. 3c, where we plot signal traces at different probe wavelengths for 662 nm pump wavelength at a carrier density of $2 \times 10^{19}$ cm$^{-3}$. For the diagonal position at 662 nm probe, we observe a rapid decay at early times after excitation. At the same time, the signal on both sides of the diagonal rises as seen by the kinetics of the 600 and 745 nm probe. This demonstrates that carriers initially excited at 662 nm scatter into other energetic states and that the carrier distribution function broadens. We plot similar dynamics in Fig. 3d for a pump wavelength of 720 nm. Again, we observe a decay of the diagonal signal and a rise at longer and shorter wavelengths. However, the timescale is now longer than for 662 nm pump. In the following, we use the decay time of the signal along the diagonal of the 2DES map as a measure for the thermalization time. From mono-exponential fits to this decay, we derive a thermalization time constant of 15 fs for 662 nm pump and 45 fs for 720 nm pump. This difference in thermalization time constants can be explained by the higher kinetic energy of carriers pumped at 662 nm, which makes the time between carrier scattering events shorter than for carriers pumped at 720 nm. It will take a few time constants for the carriers to reach a fully thermal distribution. It is, however, difficult to quantify the time point of completed carrier thermalization due to the asymptotic nature of the process.

**Thermalization time is strongly pump energy dependent**. To gain more insights on the underlying scattering process that leads to thermalization of photo-excited carriers, we study the fluence and pump wavelength dependence of the diagonal peak decay time. Figure 4a shows the dynamics of different points on the diagonal of

the 2DES maps. We observe that the thermalization time is strongly dependent on pump wavelength and measure a thermalization time constant of 85 fs for excitation near the band edge (737 nm), which decreases to sub-10-fs when exciting at 581 nm. We determine carrier scattering rates by fitting the dynamics of the diagonal peaks with an exponential decay $\propto \exp(-k_{scat}t)$. Figure 4b shows the excess energy dependence of the carrier scattering rate for the two excitation densities. In both measurements, we find an increasing scattering rate with increasing excess energy above the bandgap. When comparing the two fluences, we observe that the scattering rates are lower for the lower excitation density. The rate at which carriers scatter during thermalization is thus carrier density and excess energy dependent, indicating that the dominant thermalization process is carrier-carrier scattering.

## Discussion

2DES is an excellent tool for studying ultrafast thermalization processes with high temporal and energetic resolution. We measured thermalization time constants from below 10 to 85 fs for lead iodide perovskite depending on the excess energies of carriers. These time scales are fast compared to GaAs where carrier thermalization times have been measured in the range of 100 fs to 4 ps at room temperature[12, 15, 17]. Interestingly, Hunsche et al.[18] report no significant dependence of thermalization times on carrier density and excess energy for GaAs. The carrier thermalization times we observe for perovskites, however, show a strong dependence on both excess energy and carrier density. For an excess energy of 60 meV at an excitation density of $2 \times 10^{18}$ cm$^{-3}$, we measure a thermalization time of 70 fs (Fig. 4b). This is three times faster than the 200 fs reported for GaAs for similar excitation conditions[18]. The origin for the faster carrier-carrier scattering in hybrid perovskites is likely to be due to a weaker Coulomb screening compared to GaAs. The carrier-carrier

scattering rate $k_{e-e}$ is expected to depend on the optical (high-frequency) dielectric constant $\varepsilon$ according to $k_{e-e} \sim 1/\varepsilon^2$[31]. With a dielectric constant of 6.5–8[32, 33] for perovskite and about 11 for GaAs[34], we expect carrier-carrier scattering in perovskites to be faster by a factor of $(11/7.25)^2 = 2.3$ due to a weaker Coulomb screening. This rough estimate already gives reasonable agreement with our extracted scattering rate. However, we expect that detailed theoretical calculations, which are beyond the scope of the current report, will give more accurate values. These fast carrier scattering processes in perovskites eventually also destroy the electronic coherence of carriers, so that the thermalization times are an upper boundary for the coherence times.

Processes that can lead to carrier thermalization include carrier-carrier scattering, carrier-phonon scattering and carrier-impurity scattering. The increase of the scattering rates with increasing fluence suggests that the energy redistribution time for each carrier depends on the density of surrounding carriers. Furthermore, we observe a continuous increase of the scattering rates with excess energy and thus kinetic energy of the carriers. This suggests that the dominant scattering process for thermalization is carrier-carrier scattering under the investigated carrier densities. This can include scattering of electrons with electrons and holes with holes as well as scattering in between these two species. There might also be a contribution from carrier-phonon scattering, which would cause higher scattering rates in the hot phonon effect regime.

Carrier thermalization will ultimately limit the time for carrier extraction in hot carrier extracting solar cells (see Fig. 4c). Extraction of hot carriers has been reported for perovskite nanocrystals[35] and was recently suggested for polycrystalline lead iodide films[36]. Under the pulsed excitation regime used in these reports, hot carrier extraction is only limited by the carrier cooling time. However, under continuous illumination, such as standard sunlight illumination, there will be a large background population of cold carriers in the polycrystalline perovskite layer (around $10^{14}$–$10^{15}$ cm$^{-3}$ assuming an absorbed photon flux of around $10^{10}$ cm$^{-3}$ ps$^{-1}$) due to imperfect carrier extraction and due to the long carrier lifetimes of 100 s of nanoseconds[4, 6] compared to the cooling time of less than 1 ps[19-21]. This cold population will undergo thermalization with any newly excited charge carriers without a significant change in the temperature of the total carrier population. The excess energy will therefore be rapidly lost after carrier thermalization. Even reported longer cooling times of around 100 ps for a sub-population of the carriers[37] are still far shorter than the lifetime of charge carriers making hot carrier extraction difficult.

Strong carrier-carrier scattering can limit the carrier mobility $\mu$ under high excitation densities. By using the expression

$$\mu = \frac{e}{2m_{\text{eff}}} \cdot \tau_{\text{scat}} \qquad (2)$$

we can estimate an upper boundary for the charge carrier mobility. Here, $m_{\text{eff}}$ denotes the effective mass of the carriers ($m_{\text{eff}} \approx 0.15 m_e$ for perovskites[19, 38]), $e$ the elementary charge and $\tau_{\text{scat}}$ the average scattering time. By using the thermalization time constant near the band edge, in the range of 85 fs, we estimate upper boundaries for the mobility of 500 cm$^2$ V$^{-1}$ s$^{-1}$ for a carrier density of $2 \times 10^{19}$ cm$^{-3}$. Even with the fastest measured thermalization time constant of 8 fs, the mobility would only be limited to 50 cm$^2$ V$^{-1}$ s$^{-1}$. These values are within the range of reported carrier mobilities at low excitation densities[38]. Other carrier momentum scattering processes like acoustic phonon scattering might limit the mobilities to lower values as we did not probe carrier momentum relaxation in our experiment. We note that these estimates for the mobility are an average mobility for electron and hole, since the extracted thermalization time constant is measuring electrons and holes.

We identified the main scattering process to be carrier-carrier scattering which will get slower with lower fluence. Eventually, at low fluences, the thermalization times will be limited by carrier-impurity scattering and carrier-phonon scattering. The latter has been shown to occur on timescales of 200–400 fs[19, 20], in agreement with our measurements.

In conclusion, we report on carrier thermalization in lead iodide perovskite measured by 2DES. We find thermalization time constants of below 10–85 fs with carrier-carrier scattering being the dominant process. Furthermore, we discussed that these timescales are the limiting factor for hot carrier extracting devices. The reported timescales give an insight into the fundamental carrier–carrier interactions and provide a deeper understanding of the photophysics of these emerging photovoltaic materials.

## Methods

**Film preparation**. For the iodide perovskite films, 3:1 molar stoichiometric ratios of $CH_3NH_3I$ and $Pb(CH_3COO)_2$ (Sigma Aldrich 99.999% pure) were made in $N,N$-dimethylformamide in 20 wt% solution. This solution was spun inside a nitrogen filled glove box on quartz substrates at 2000 r.p.m. for 60 s followed by 3 min of thermal annealing at 100 °C in air to form thin films. The samples were encapsulated with a second glass slide and epoxy adhesive (Loctite Double Bubble) under inert conditions to avoid sample degradation and beam damage.

**Pump–probe experiment**. We perform pump–probe experiments with sub-10 fs laser pulses. The pump–probe setup starts with an amplified Ti:sapphire laser system (Libra, Coherent), which delivers 4-mJ, 100-fs pulses around 800 nm at 1-kHz repetition rate. A portion of the laser with 300 μJ energy is used to pump a non-collinear optical parametric amplifier (NOPA), which is subsequently split into pump and probe pulse. The NOPA delivers a pulse with a spectrum spanning from 550 nm (2.25 eV) to 750 nm (1.65 eV), as shown in Fig. 1a, compressed to sub-10-fs duration by multiple reflections on custom-designed double-chirped mirrors (DCMs). Pulse duration is measured by second harmonic generation frequency resolved optical gating[39]. The pump energy is 3 nJ which, focused to a spot size of $\approx 100$ μm, yields a fluence of $\approx 10$ μJ cm$^{-2}$.

**Two-dimensional electron spectroscopy**. We perform 2DES in the partially collinear pump–probe geometry, according to the scheme shown in Supplementary Fig. 2. 2DES can be seen as an extension of conventional pump–probe spectroscopy, where two identical collinear pump pulses are used and their delay $t_1$ (coherence time) is scanned in time, for a fixed value of the probe pulse delay $t_2$ (population time). The probe pulse is dispersed in a spectrometer, providing resolution in the detection frequency. The FT with respect to the pump pulses delay provides the resolution of the signals with respect to the excitation frequency[40-42].

The 2DES setup uses the same NOPA as pump–probe, which is divided by a beam splitter (90% transmission, 10% reflection) into pump and probe lines. The identical and phase-locked pair of femtosecond pump pulses is generated by the Translating-Wedge-Based Identical-Pulses-eNcoding System (TWINS) technology[43, 44]. TWINS uses birefringence to impose user-controlled temporal delays, with attosecond precision, between two orthogonal components of broadband laser pulses. Rapid scanning of the inter-pulse delay allows robust and reliable generation of 2DES spectra in a user-friendly pump–probe geometry. In order to determine zero delay between the pump pulses and properly phase the 2DES spectra, part of the pump beam is split off and sent to a photodiode to monitor the interferogram of the pump pulse pair. The additional dispersion introduced by the TWINS on the pump pulse pair is compensated by a suitable number of bounces on a pair of DCMs, and spectral phase correction is verified using a Spatially Encoded Arrangement for Temporal Analysis by Dispersing a Pair Of Light E-fields (SEA-TADPOLE) setup[45]. Pump and probe pulses are non-collinearly focused on the sample and the transient transmission change $\Delta T/T$ is measured by a spectrometer[46].

**Data availability**. The experimental data that support the findings of this study are available in the University of Cambridge Repository (https://doi.org/10.17863/CAM.11883).

## References

1. Weber, D. $CH_3NH_3PbX_3$, ein Pb(II)-System mit kubischer Perowskitstruktur/ $CH_3NH_3PbX_3$, a Pb(II)-system with cubic perovskite structure. *Z. Naturforsch. B* **33**, 1443–1445 (1978).

# ARTICLE

2. Lee, M. M., Teuscher, J., Miyasaka, T., Murakami, T. N. & Snaith, H. J. Efficient hybrid solar cells based on meso-superstructured organometal halide perovskite. *Science* **338**, 643–647 (2012).

3. Kim, H.-S. et al. Lead iodide perovskite sensitized all-solid-state submicron thin film mesoscopic solar cell with efficiency exceeding 9%. *Sci. Rep.* **2**, 591 (2012).

4. Dong, Q. et al. Electron-hole diffusion lengths 175 um in solution-grown $CH_3NH_3PbI_3$ single crystals. *Science* **347**, 967–970 (2015).

5. Protesescu, L. et al. Nanocrystals of cesium lead halide perovskites ($CsPbX_3$, X=Cl, Br, and I): novel optoelectronic materials showing bright emission with wide color gamut. *Nano Lett.* **15**, 3692–3696 (2015).

6. Wehrenfennig, C., Eperon, G. E., Johnston, M. B., Snaith, H. J. & Herz, L. M. High charge carrier mobilities and lifetimes in organolead trihalide perovskites. *Adv. Mater.* **26**, 1584–1589 (2014).

7. Sadhanala, A. et al. Preparation of single-phase films of $CH_3NH_3Pb(I_{1-x}Br_x)_3$ with sharp optical band edges. *J. Phys. Chem. Lett.* **5**, 2501–2505 (2014).

8. De Wolf, S. et al. Organometallic halide perovskites: sharp optical absorption edge and its relation to photovoltaic performance. *J. Phys. Chem. Lett.* **5**, 1035–1039 (2014).

9. de Quilettes, D. W. et al. Impact of microstructure on local carrier lifetime in perovskite solar cells. *Science* **348**, 683–686 (2015).

10. Richter, J. M. et al. Enhancing photoluminescence yields in lead halide perovskites by photon recycling and light out-coupling. *Nat. Commun.* **7**, 13941 (2016).

11. Deschler, F. et al. High photoluminescence efficiency and optically pumped lasing in solution-processed mixed halide perovskite semiconductors. *J. Phys. Chem. Lett.* **5**, 1421–1426 (2014).

12. Oudar, J. L., Hulin, D., Migus, A., Antonetti, A. & Alexandre, F. Subpicosecond spectral hole burning due to nonthermalized photoexcited carriers in GaAs. *Phys. Rev. Lett.* **55**, 2074–2077 (1985).

13. Brito Cruz, C. H., Gordon, J. P., Becker, P. C., Fork, R. L. & Shank, C. V. Dynamics of spectral hole burning. *IEEE J. Quantum Electron.* **24**, 261–269 (1988).

14. Shah, J. *Ultrafast Spectroscopy of Semiconductors and Semiconductor Nanostructures* vol. 115 of *Springer Series in Solid-State Sciences* (Springer, 1999).

15. Schoenlein, R. W., Lin, W. Z., Ippen, E. P. & Fujimoto, J. G. Femtosecond hot-carrier energy relaxation in GaAs. *Appl. Phys. Lett.* **51**, 1442–1444 (1987).

16. Rota, L., Lugli, P., Elsaesser, T. & Shah, J. Ultrafast thermalization of photoexcited carriers in polar semiconductors. *Phys. Rev. B* **47**, 4226–4237 (1993).

17. Elsaesser, T., Shah, J., Rota, L. & Lugli, P. Initial thermalization of photoexcited carriers in GaAs studied by femtosecond luminescence spectroscopy. *Phys. Rev. Lett.* **66**, 1757–1760 (1991).

18. Hunsche, S., Heesel, H., Ewertz, A., Kurz, H. & Collet, J. H. Spectral-hole burning and carrier thermalization in GaAs at room temperature. *Phys. Rev. B* **48**, 17818–17826 (1993).

19. Price, M. B. et al. Hot-carrier cooling and photoinduced refractive index changes in organic–inorganic lead halide perovskites. *Nat. Commun.* **6**, 8420 (2015).

20. Yang, Y. et al. Observation of a hot-phonon bottleneck in lead-iodide perovskites. *Nat. Photonics* **10**, 53–59 (2015).

21. Chen, K., Barker, A. J., Morgan, F. L. C., Halpert, J. E. & Hodgkiss, J. M. Effect of carrier thermalization dynamics on light emission and amplification in organometal halide perovskites. *J. Phys. Chem. Lett.* **6**, 153–158 (2015).

22. Piatkowski, P. et al. Direct monitoring of ultrafast electron and hole dynamics in perovskite solar cells. *Phys. Chem. Chem. Phys.* **17**, 14674–14684 (2015).

23. Yang, J. et al. Acoustic-optical phonon up-conversion and hot-phonon bottleneck in lead-halide perovskites. *Nat. Commun.* **8**, 14120 (2017).

24. Brida, D. et al. Ultrafast collinear scattering and carrier multiplication in graphene. *Nat. Commun.* **4**, 1987 (2013).

25. Jonas, D. M. Two-dimensional femtosecond spectroscopy. *Annu. Rev. Phys. Chem.* **54**, 425–463 (2003).

26. Miyata, A. et al. Direct measurement of the exciton binding energy and effective masses for charge carriers in organic–inorganic tri-halide perovskites. *Nat. Phys.* **11**, 582–587 (2015).

27. Fluegel, B. et al. Femtosecond studies of coherent transients in semiconductors. *Phys. Rev. Lett.* **59**, 2588–2591 (1987).

28. Sokoloff, J. P. et al. Transient oscillations in the vicinity of excitons and in the band of semiconductors. *Phys. Rev. B* **38**, 7615–7621 (1988).

29. Kobayashi, T. et al. Real-time spectroscopy of single-walled carbon nanotubes for negative time delays by using a few-cycle pulse laser. *J. Phys. Chem. C* **118**, 3285–3294 (2014).

30. Yan, S., Seidel, M. T. & Tan, H.-S. Perturbed free induction decay in ultrafast mid-IR pump–probe spectroscopy. *Chem. Phys. Lett.* **517**, 36–40 (2011).

31. Del Fatti, N. et al. Nonequilibrium electron dynamics in noble metals. *Phys. Rev. B* **61**, 16956–16966 (2000).

32. Hirasawa, M., Ishihara, T., Goto, T., Uchida, K. & Miura, N. Magnetoabsorption of the lowest exciton in perovskite-type compound $(CH_3NH_3)PbI_3$. *Phys. B Condens. Matter* **201**, 427–430 (1994).

33. Lin, Q., Armin, A., Nagiri, R. C. R., Burn, P. L. & Meredith, P. Electro-optics of perovskite solar cells. *Nat. Photonics* **9**, 106–112 (2015).

34. Madelung, O., Rössler, U. & Schulz, M. (eds). *Group IV Elements, IV-IV and III-V Compounds. Part a—Lattice Properties* 1–11 (Springer-Verlag, 2001).

35. Li, M. et al. Slow cooling and highly efficient extraction of hot carriers in colloidal perovskite nanocrystals. *Nat. Commun.* **8**, 14350 (2017).

36. Guo, Z. et al. Long-range hot-carrier transport in hybrid perovskites visualized by ultrafast microscopy. *Science* **356**, 59–62 (2017).

37. Zhu, H. et al. Screening in crystalline liquids protects energetic carriers in hybrid perovskites. *Science* **353**, 1409–1413 (2016).

38. Brenner, T. M. et al. Are mobilities in hybrid organic–inorganic halide perovskites actually "High"? *J. Phys. Chem. Lett.* **6**, 4754–4757 (2015).

39. Trebino, R. *Frequency-Resolved Optical Gating: The Measurement of Ultrashort Laser Pulses* (Springer, 2000).

40. Mukamel, S. *Principles of Nonlinear Optical Spectroscopy* (Oxford University Press, 1995).

41. Brańczyk, A. M., Turner, D. B. & Scholes, G. D. Crossing disciplines—a view on two-dimensional optical spectroscopy. *Ann. Phys.* **526**, 31–49 (2014).

42. Hamm, P. & Zanni, M. *Concepts and Methods of 2D Infrared Spectroscopy* (Cambridge University Press, 2011).

43. Réhault, J., Maiuri, M., Oriana, A. & Cerullo, G. Two-dimensional electronic spectroscopy with birefringent wedges. *Rev. Sci. Instrum.* **85**, 123107 (2014).

44. Brida, D., Manzoni, C. & Cerullo, G. Phase-locked pulses for two-dimensional spectroscopy by a birefringent delay line. *Opt. Lett.* **37**, 3027–3029 (2012).

45. Bowlan, P. et al. Crossed-beam spectral interferometry: a simple, high-spectral-resolution method for completely characterizing complex ultrashort pulses in real time. *Opt. Express* **14**, 11892–11900 (2006).

46. Polli, D., Lüer, L. & Cerullo, G. High-time-resolution pump-probe system with broadband detection for the study of time-domain vibrational dynamics. *Rev. Sci. Instrum.* **78**, 103108 (2007).

## Acknowledgements

This project has received funding from the European Union's Horizon 2020 research and innovation programme under grant agreement no 654148 Laserlab-Europe (CUSBO 002151). We acknowledge further financial support from the Engineering and Physical Sciences Research Council of the UK (EPSRC). G.C. acknowledges support by the European Union Horizon 2020 Programme under Grant Agreement No. 696656 Graphene Flagship and by the European Research Council Advanced Grant STRATUS (ERC-2011-AdG No. 291198). J.M.R. and F.D. thank the Winton Programme for the Physics of Sustainability (University of Cambridge). J.M.R. thanks the Cambridge Home European Scheme for financial support. F.D. acknowledges funding from a Herchel Smith Research Fellowship and a Winton Advanced Research Fellowship. We thank Cristian Manzoni for fruitful discussions.

## Author contributions

J.M.R., F.D. and G.C. conceived the experiment. B.Z. and J.M.R. prepared the samples. J.M.R., F.B. and F.V.d.A.C. performed the experiments. J.M.R., F.B. and F.D. analyzed the data. All authors discussed the data and contributed to the manuscript.

## Additional information

**Competing interests:** The authors declare no competing financial interests.

