## [Peer Review File · Nature Communications]

Reviewers' comments:

Reviewer #1 (Remarks to the Author):

The manuscript reports the time scales of ultrafast carrier thermalization in a MAPbI₃ thin film caused by carrier-carrier scattering. Using two-dimensional electronic spectroscopy (2D-ES), the authors suggested much faster intraband thermalization (8 – 85 fs) than that in conventional polar semiconductor (e.g. 100s to 4 ps in GaAs). The problem has been probed by many authors using various techniques. The use of 2D-ES is an interesting addition to the extension photophysical studies on MAPbI₃. However, I am highly skeptical of the authors' analysis and conclusions. The analysis on the short time scale is particularly problematic and likely wrong. Both the novelty and the broad impact are also limited.

1) The MAPbI₃ thin film sample is very sensitive to preparation conditions and easily damaged under photo-excitation. The authors need to provide details on sample characterization. Given the laser excitation density used, degradation of the sample is a major concern.

2) The authors claim that the spectra of the output from NOPA span the range of 550 – 800 nm, but the spectrum seen in Fig. 1 does not look like having enough intensity around 750 – 800 nm. This leads natural question against the reliability of the data in this region, which is actually very important region because bleaching from exciton resonance exists.

3) The authors did not take into the oscillations observed at negative time delay, such a coherent artifacts can distort the important time scale especially as short as instrumental response. In particular, I am concerned that such coherent artifacts can have major effect on the analysis of short time data. This is a serious problem as the authors put too much emphasis on the ultrafast time scale (8-85 fs).

4) The Lorentzian fits to extract homogeneous widths are highly questionable. The author fit only the vicinity region to the peak, which can cause significant errors on resultant fitting parameters. In particular, y-axis offsets are unacceptably large (-2.2 and -1.2 in Fig. S4a and S4b, respectively), which makes me suspicious. At very least, the author should present larger energy region to show the reliability of the fittings. Also, it might not be very clear if the line shape should be lorentian in the case of ultrafast intraband relaxation concurrently happening like in this case.

5) The region above band gap could contain signals from multiple origins – e.g. induced absorption due to transient band renormalization. Therefore, the authors need to pay close attention to eliminate possible artifact that can lead to significant errors on the analysis thermalization timescales.

6) As for the argument on mobility, I don't think it make sense to use the scattering time of hot carriers since the mobility measured so far was for the carriers at bandedges. It make more sense for me to compare the calculated mobility from the lowest excitation range (>750 nm).

7) I would expect to see the discussion about possible origin of such a high carrier-carrier scattering compared to GaAs in such a high impact journal like Nature Communication. Possibly there is a correlation to the highly dynamic nature of lead-halide perovskites?

8) I guess the measurement detects both hole and electrons. Since it is expected that the hole and electron in the perovskites have different mobilities, interpretation of the results might need be amended.

9) For estimation of thermalization time discussed in Fig. 3, picking the trace at one particular probe wavelength can mislead the timescales. Instead, the authors might be able to show the spectral evolution from the beginning to convergence into Boltzmann like distribution.

10) In addition, I don't understand why the authors can define thermalization time as a function of excess energy. The author extract the time from simple exponential fit of the data at particular point, but the definition of thermalization should be defined as the time needed to reach Boltzmann distribution, which we need to judge from spectrum shape instead of seeing the signal intensity at certain point. Also, is there any particular reason why many points are deleted in Fig.4 (e.g. 190 – 230 meV for 2×10^{18})?

Reviewer #2 (Remarks to the Author):

Dear Editor: The manuscript reports on the use of two-dimensional electronic spectroscopy with sub-10 fs resolution to interrogate the ultrafast events in a MAPbI₃ perovskite film. The major claim is to extract the times of carrier thermalization (8-85 fs) due to carrier-carrier scattering, while the obtained cooling time is already known (hundreds of fs). The information is of interest to understand the opto-electronics (and Photophysics) of this kind of materials, and therefore to the community working in the field. The used technique is powerful, and the topics is hot. The result could be of more interest to the community using ultrafast spectroscopy and high level of theory to elucidate the photophysics of perovskite and this kind of materials. However, I have few comments and some doubt about the analysis of the data.

I suggest accepting a revised version for further review.

Below is my suggestion.

1- Will be good for no experts in ultrafast spectroscopy to better describe and in more detail (in SI part) the experiment: How the value of t_1 and the overlap between both pumping pulses are taken into account in the transient signal when you change t_2 ?

2- When both pumping pulses overlap (in time), is the resulted fluence still enough weak not to induce higher order phenomena ? It is important to check that under the used fluences of both pumps ($2 \times 10^{19} \text{ cm}^{-3}$ is too high) you do not see a such behavior.

3- For the cooling time, fluence and wavelength dependence, see also: Phys. Chem. Chem. Phys., 2015, 17, 14674—14684.

4- Be sure that the oscillation observed at around zero time is not affecting the time value of the rising component: It's the heart of the ms. One way is to "deconvolute" or analyze their transient signal using for example, the procedure shown in Chem. Phys., 200 (1959 415-429, and ref. 13. Using the described procedures there, one can have transients free from perturbed induction decay, which appears at times shorter or equal to 0 fs.

5- Why the signal in Fig. 4a, at 726 nm (pump-probe) and at time below 0 is positive and different from those of others (in the same figure).

Minor changes:

6- Give few references for equation (1).

7- Fig. 2: Delete the "pump arrows and E_{pump} " in the carton of Hot population and Cold population. We do not pump any more during these processes.

8- For film preparation, check that Pb(CH₃COO)₂ is really 99.999% ! What is the purity of DMF ?

9- The movies are not uploaded.

Reviewers' comments:

Reviewer #1 (Remarks to the Author):

The manuscript reports the time scales of ultrafast carrier thermalization in a MAPbI₃ thin film caused by carrier-carrier scattering. Using two-dimensional electronic spectroscopy (2D-ES), the authors suggested much faster intraband thermalization (8 – 85 fs) than that in conventional polar semiconductor (e.g. 100s to 4 ps in GaAs). The problem has been probed by many authors using various techniques.

We agree that carrier thermalization processes have been probed for GaAs, and these experiments have provided crucial information. Here we measure the thermalization times for hybrid lead iodide perovskite for the first time. We emphasize that, while carrier cooling dynamics in perovskites due to electron-phonon scattering has been probed in a number of studies, the primary carrier relaxation process, due to electron-electron scattering, towards a Fermi-Dirac distribution is essentially unexplored. As we describe in the manuscript, this was a significant experimental challenge due to the fast nature of the process in perovskites, which required methods beyond those used for GaAs, in particular 2D-ES that combines high temporal and spectral resolution.

The use of 2D-ES is an interesting addition to the extension photophysical studies on MAPbI₃. However, I am highly skeptical of the authors' analysis and conclusions. The analysis on the short time scale is particularly problematic and likely wrong. Both the novelty and the broad impact are also limited.

As we will discuss in greater detail in the following replies to the referee's detailed questions, we are convinced about the robustness and the validity of our analysis, also at short time scales.

In addition, and in agreement with reviewer 2, we believe that this work will have a big impact on the community studying the photophysics and fundamental carrier interactions in perovskites. We provide the first experimental study of the primary energy re-distribution process, occurring through electron-electron interactions, in these novel materials of high fundamental and technological interest. We are further convinced that the impact of our work extends well beyond the field of perovskite photophysics. To our knowledge, 2D spectroscopy has never been used before for studying carrier thermalization in semiconductors. We therefore have demonstrated a new spectroscopic tool of general validity which can be used to study thermalization processes in similar systems with band-like character.

1) The MAPbI₃ thin film sample is very sensitive to preparation conditions and easily damaged under photo-excitation. The authors need to provide details on sample characterization. Given the laser excitation density used, degradation of the sample is a major concern.

We thank the reviewer for this comment. We added to the SI a plot of the 2DES signal over experimental time under high excitation density ($2 \times 10^{19} \text{ cm}^{-3}$). We observe that no significant sample degradation took place over several hours of measurement time, indicating that our prepared samples were of high quality and that our excitation densities were below the damage threshold (see Fig. R1). All measurements were performed on samples encapsulated in a nitrogen atmosphere, thus prevented degradation through interaction with atmospheric oxygen. For sample fabrication, we followed standard procedures, as now described in the Methods section. We extended our preparation description in the Methods section (page 10, line 13 – 19).

Figure R1: Sample stability. 2DES signal plotted over the sweep number where one sweep takes about 13 minutes. We observe no significant sample degradation over many hours of experimental time.

2) The authors claim that the spectra of the output from NOPA span the range of 550 – 800 nm, but the spectrum seen in Fig. 1 does not look like having enough intensity around 750 – 800 nm. This leads natural question against the reliability of the data in this region, which is actually very important region because bleaching from exciton resonance exists.

We thank the reviewer for pointing out this inaccuracy. We changed these numbers in the manuscript text to ‘from 550 nm to 750 nm’ (page 3, line 29). The low light intensity between 750 nm and 800 nm means that we don’t pump many carriers in this region. The light intensity is, however, still strong enough to probe this spectral region, since pump-probe signals are independent of the probe intensity. For studying carrier thermalization, we are mainly interested in looking at free carriers above the bandgap and we do not focus specifically on the signal from the region close to the excitonic resonance, which will be addressed in a follow-up study. Our broadband excitation pulses (550-750 nm) allow us to study free carrier thermalization up to 500 meV above the bandgap. In the future using a different excitation pulse (700-900 nm bandwidth, see Opt. Lett. **34**, 3592 (2009)) we plan to address the excitonic transition with the same methodologies employed in this paper.

3) The authors did not take into the oscillations observed at negative time delay, such a coherent artifacts can distort the important time scale especially as short as instrumental response. In particular, I am concerned that such coherent artifacts can have major effect on the analysis of short time data. This is a serious problem as the authors put too much emphasis on the ultrafast time scale (8-85 fs).

We acknowledge that our previous manuscript was not clear enough regarding the nature of the coherent oscillations at negative times and their relationship with the coherent artefact. We would like to point out that these oscillations are expected to only occur at negative times (before temporal overlap of pump and probe pulse) and thus will not affect our analysis of thermalization at positive times. As detailed in the new section in the SI, the oscillations at negative times (sometimes

referred to as the “pump perturbed free-induction decay” or PP-FID) arise because inversion of the time-ordering of pump and probe alters the phase-matching of the setup. In this configuration, two double-sided Feynman diagrams with one interaction with the probe and two interactions with the pump emit a signal along the probe’s propagation direction.

We emphasize that the time ordering of the pulses affects the phase matching of the experiment and consequently which pathways will be detected. However, causality is always respected. Therefore, all pathways observed at positive delays generate no signal at negative delays, and vice-versa. Therefore, the signal observed at positive delays, which is the one in which we are interested in order to monitor carrier thermalization, is not affected by the oscillating signal at negative delays due to PP-FID. We believe that the expanded SI suffices to point non-specialist readers to the heart of the issue and to helpful literature where it is detailed.

The referee is correct to point out that in our analysis we do not attempt to model the coherent artefacts occurring around time zero. Such nonlinear signals arise partially from the presence of many extra pathways when there is pulse overlap and therefore the time ordering of the pulses is not well-defined, as well as from the non-resonant nonlinear response from the sample and the substrate. Because of the highly nonlinear nature of the coherent artefacts, there is no robust way to model them, although sophisticated tools such as global analysis can be used to try to unravel fast decays from the coherent artefact. We emphasize that the duration of the coherent artefacts is of the order of the instrument response function (IRF) of the setup which, in our degenerate configuration, corresponds to the autocorrelation of the 7-fs excitation pulses. Thus, although it is true that the thermalization times approach our 7 fs pulse duration for the fastest extracted thermalization decay fits, most of the extracted thermalization times are well above this time resolution, so that the thermalization dynamics are not affected by the coherent artefact, as clearly shown by the dynamics in Figs. 3c, 3d and 4a. We stress that the importance of our results lies in the clear trends observed for the thermalization timescale as a function of the original excess energy of the carriers, both in the low and high fluence regimes. In order to improve the manuscript’s clarity on this point, we have worded the results differently (e.g. page 4, line 13 – 14, page 8, line 9) and added error bars to figure 4b to quantify the uncertainties in the extracted thermalization timescales.

4) The Lorentzian fits to extract homogeneous widths are highly questionable. The author fit only the vicinity region to the peak, which can cause significant errors on resultant fitting parameters. In particular, y-axis offsets are unacceptably large (-2.2 and -1.2 in Fig. S4a and S4b, respectively), which makes me suspicious. At very least, the author should present larger energy region to show the reliability of the fittings. Also, it might not be very clear if the line shape should be lorentian in the case of ultrafast intraband relaxation concurrently happening like in this case.

We agree with the reviewer that the analysis of the lineshape is not straightforward and that our modelling of the lineshapes as Lorentzian, which strictly applies to a two-level system, is not straightforward to extend to a semiconductor above bandgap. We plot below the lineshapes over a broader energy range, as suggested by the reviewer. We find that these peaks are overlapping with negative signals on both sides of the peak, as described in the manuscript and as reported for GaAs before. This makes the exact fitting of the peak width difficult. We therefore decided to remove the analysis of these peak widths from our manuscript, as we think that it is not of great importance for our report on thermalization times. However, the supplementary 2D movies clearly show how the diagonal peak width qualitatively increases at higher fluences when the thermalization times

become faster. This indicates that the reported thermalization times and observed peak widths are qualitatively in good agreement.

Figure R2: Homogeneous broadening. Antidiagonal width of diagonal peak at time zero at (A) 645 nm and (B) 737 nm. The peak is extracted perpendicular to the diagonal line in the 2D map and fitted with a Lorentzian function.

5) The region above band gap could contain signals from multiple origins – e.g. induced absorption due to transient band renormalization. Therefore, the authors need to pay close attention to eliminate possible artifact that can lead to significant errors on the analysis thermalization timescales.

We agree that various effects can give rise to a differential transmission ($\Delta T/T$) signal in perovskites. The mentioned transient band renormalization is important for the below-bandgap region (760-800 nm, see Price et al., Nat. commun., 2015, 6 8420), but less important in the above-bandgap region, since the absorption spectrum is rather flat in this region. More important for the above-bandgap region is a transient reflectivity change which causes a negative $\Delta T/T$ signal (Price et al., Nat. commun., 2015, 6 8420) as mentioned in the manuscript (page 4, line 5 – 6 and page 7, line 7 – 12). However, this overlapping negative signal does not affect the extracted thermalization times.

6) As for the argument on mobility, I don't think it make sense to use the scattering time of hot carriers since the mobility measured so far was for the carriers at bandedges. It make more sense for me to compare the calculated mobility from the lowest excitation range (>750 nm).

We agree with the reviewer, and changed the used thermalization time to 85 fs which is the value that we measure at 737 nm pump wavelength and thus close to the bandedge.

7) I would expect to see the discussion about possible origin of such a high carrier-carrier scattering compared to GaAs in such a high impact journal like Nature Communication. Possibly there is a correlation to the highly dynamic nature of lead-halide perovskites?

We thank the reviewer for this comment. As requested, we added a discussion of the origin of such fast thermalization times to the paper (page 8, line 17 – 25). The faster thermalization is likely to be due to a weaker screening of the Coulomb interaction in perovskites compared to GaAs due to the lower dielectric constant of perovskites ($\epsilon \approx 7.25$) with respect to GaAs ($\epsilon \approx 11$). This

reduced screening qualitatively accounts very well for the increased carrier-carrier scattering rate observed in perovskites.

8) I guess the measurement detects both hole and electrons. Since it is expected that the hole and electron in the perovskites have different mobilities, interpretation of the results might need be amended.

We agree with the reviewer that we cannot distinguish between electrons and holes. We added a discussion of this to the manuscript (page 9, line 5 – 6 and line 27 – 29). Since we are measuring the thermalization of both electrons and holes, the derived upper boundary for the mobility is an average boundary.

In this section, we also add a detailed discussion of the importance of thermalization times on the performance of hot carrier devices (page 9, line 7 – 17). We believe, this discussion will be very helpful for future development of perovskite hot carrier devices and increases the impact of our results.

9) For estimation of thermalization time discussed in Fig. 3, picking the trace at one particular probe wavelength can mislead the timescales. Instead, the authors might be able to show the spectral evolution from the beginning to convergence into Boltzmann like distribution.

We stress that the thermalization times were extracted from the decay of the diagonal signal in Figure 4a, not in Figure 3.

As suggested by the reviewer, we added a sub-figure (Figure 3b) which shows the spectral evolution from a peak at $E_{\text{pump}} = E_{\text{probe}}$ at 0fs to a Boltzmann distribution at 100fs.

10) In addition, I don't understand why the authors can define thermalization time as a function of excess energy. The author extract the time from simple exponential fit of the data at particular point, but the definition of thermalization should be defined as the time needed to reach Boltzmann distribution, which we need to judge from spectrum shape instead of seeing the signal intensity at certain point. Also, is there any particular reason why many points are deleted in Fig.4 (e.g. 190 – 230 meV for 2×10^{18})?

We agree that the thermalization time is the time required for the carrier population to reach a Boltzmann distribution. As shown in Figure 3c and 3d, we observe that the rise of the $\Delta T/T$ signal at higher and lower detection energies occurs on the same timescale as the decay of the signal at the excitation energy, because carriers excited at a given energy quickly redistribute their energy. We can therefore define the decay time of the diagonal signal, corresponding to $E_{\text{pump}} = E_{\text{probe}}$, as the thermalization time (see page 7, line 16 – 17). This is a particularly strong definition as it avoids any subjective judgment of whether or not a carrier distribution is Boltzmann-like.

In Figure 4b, the data points were only plotted for energies where the signal-to-noise ratio was sufficient to obtain a good fit of the diagonal decay. It happened that this was more challenging for the low-fluence measurement, due to the lower signal, than for the high-fluence experiment, which reduced the number of available data points in the former case.

Reviewer #2 (Remarks to the Author):

Dear Editor: The manuscript reports on the use of two-dimensional electronic spectroscopy with sub-10 fs resolution to interrogate the ultrafast events in a MAPbI₃ perovskite film. The major claim is to extract the times of carrier thermalization (8-85 fs) due to carrier-carrier scattering, while the obtained cooling time is already known (hundreds of fs). The information is of interest to understand the opto-electronics (and Photophysics) of this kind of materials, and therefore to the community working in the field. The used technique is powerful, and the topics is hot. The result could be of more interest to the community using ultrafast spectroscopy and high level of theory to elucidate the photophysics of perovskite and this kind of materials.

However, I have few comments and some doubt about the analysis of the data. I suggest accepting a revised version for further review.

We thank the reviewer for his/her evaluation of the importance of our work, both in terms of the addressed topic (which is “hot”) and of the used spectroscopic technique (which is “powerful”). We are also convinced that our results, shedding light on the so far unexplored thermalization timescale in perovskites, will be of interest for the broad community working on opto-electronic applications of these materials. In the following we fully address the reviewer’s comments.

Below is my suggestion.

1- Will be good for no experts in ultrafast spectroscopy to better describe and in more detail (in SI part) the experiment: How the value of t_1 and the overlap between both pumping pulses are taken into account in the transient signal when you change t_2 ?

We added a more detailed description of the experimental setup in the SI, introducing the concept of the two-dimensional electronic spectroscopy experiment and discussing how we achieved the necessary requirements.

2- When both pumping pulses overlap (in time), is the resulted fluence still enough weak not to induce higher order phenomena ? It is important to check that under the used fluences of both pumps ($2 \times 10^{19} \text{ cm}^{-3}$ is too high) you do not see a such behavior.

We thank the reviewer for this comment. We added a graph to the Supplementary Information where we plot the strength of the photo-bleaching signal as a function of the excitation laser fluence. We observe that the photo-bleaching signal is proportional to the exciting laser fluence, indicating that we are dominantly in the linear excitation regime. We therefore assume that higher order processes play a minor role in our range of excitation fluences.

Figure R3: Fluence dependence of the $\Delta T/T$ signal. $\Delta T/T$ signal plotted as a function of excitation fluence (black squares) with a linear fit (blue line). The signal is proportional to the excitation energy indicating that we are measuring in a linear excitation regime and that higher order absorption plays a minor role under these conditions.

3- For the cooling time, fluence and wavelength dependence, see also: Phys. Chem. Chem. Phys., 2015, 17, 14674—14684.

We thank the reviewer for guiding us to this publication. We added it to our discussion of the relevant literature about carrier cooling (page 3, line 1 – 4).

4- Be sure that the oscillation observed at around zero time is not affecting the time value of the rising component: It's the heart of the ms. One way is to "deconvolute" or analyze their transient signal using for example, the procedure shown in Chem. Phys., 200 (1959 415-429, and ref. 13). Using the described procedures there, one can have transients free from perturbed induction decay, which appears at times shorter or equal to 0 fs.

We agree with the reviewer that the interpretation of negative times is very challenging. We also stress that we are not using the rise of the signal at negative times in any part of our analysis. Due to the problems raised by the reviewer, we limit our analysis to positive time delays where no oscillations are expected and where carrier thermalization is occurring. As mentioned in the reply to comment 3 by reviewer 1, disentangling the decay time from the coherent artefact is not the main goal of the paper, or required to obtain reliable carrier scattering rates. Instead, the observed trend in thermalization times as a function of fluence and excess energy is the main result.

The reviewer also asks about the signal at negative times (pump perturbed free-induction decay), as the paper cited by the reviewer is a classic work by Peter Hamm addressing potential confusion of the signal at negative times with real signal in time resolved IR spectroscopy. We stress once more that this signal is identically zero following the pulse overlap region, so the only contribution that arises from the instrument at positive times is the usual coherent artefact during pulse overlap (approximately 7 fs in our case). We hope that the added discussion in the SI on the experimental method helps to clarify the issue.

5- Why the signal in Fig. 4a, at 726 nm (pump-probe) and at time below 0 is positive and different from those of others (in the same figure).

This difference might be due to the fact that at 726 nm we are already probing a tail of the excitonic transition. In this case the signal at negative times rises over a timescale comparable to the dephasing time of the excitonic transition (see e.g. T. Guenther *et al.*, Phys. Rev. Lett. **89**, 057401 (2002)), which can be different from the dephasing times of the carrier continuum region.

However, the interpretation of the 2D signal at negative time delays is challenging due to a change in the pulse sequence of pump and probe pulses. We thus don't draw any conclusions from the 2D data at negative time delays.

Minor changes:

6- Give few references for equation (1).

Due to concerns from reviewer 1 regarding the interpretation of the peak width, we decided to remove this part from our manuscript as we think it is of low importance to our report which focuses on the observation of carrier thermalization.

7- Fig. 2: Delete the "pump arrows and E_{pump} " in the cartoon of Hot population and Cold population. We do not pump any more during these processes.

We thank the reviewer for pointing this out. The cartoon has been changed accordingly.

8- For film preparation, check that $\text{Pb}(\text{CH}_3\text{COO})_2$ is really 99.999% ! What is the purity of DMF ?

We used $\text{Pb}(\text{CH}_3\text{COO})_2$ from Sigma-Aldrich (316512) with a purity of 99.999%. This is reflected by the high external photoluminescence quantum efficiency (PLQE) of films prepared for reference measurements, which reached ~15% external PLQE.

9- The movies are not uploaded.

We are very sorry that there was a problem when uploading the movies. We sent them directly to the editor when we realised and they should now be available.

Reviewers' comments:

Reviewer #1 (Remarks to the Author):

While the revision addressed some of the questions, my major concerns remain.

1. My first concern is still on data analysis. The experimental approach was powerful, but the analysis and interpretation on several time scales and corresponding scattering processes got mixed up. Let me go into details.

(1) Momentum scattering. This is what enters into the mobility in the Drude model. The Momentum scattering rate cannot be extracted from a 2D-TA experiment, since TA does not give momentum resolution. I agree with the authors, though, that the scattering rate observed in the experiment gives a lower limit for the momentum scattering rate. The difference between the two should nevertheless be made clear.

(2) Energy relaxation rate. This is what the authors measure in their analysis of the data by taking cuts at selected probe energies.

(3) Thermalization time. It takes a large number of energy and momentum scattering events for the electrons to reach a thermal distribution. Taking a look at Figure 3b, I would say that - if at all within the time window shown - a thermalized distribution is reached after 100 fs at the best. Attributing the curves in Fig. 3 c and d or the cross matrix elements in Fig. 4 to the thermalization time is speculative. What one can see from those graphs is that the initial relaxation of the excess energy takes place on the time scale extracted. This does not give any information about whether the resulting carrier distribution is a thermalized one, which is a much more complicated question to answer. Thermalization times need to be extracted from plots like in Fig. 3b or S5. More importantly, cooling times need to be extracted from plots like in Fig. 3b, for example by analyzing the average energy or the high-energy tail of the distribution. Temperature (and average energy, which is often used as a measure of temperature) is an ensemble property, and the analysis in Fig. S6 does not give this information.

(4) Cooling time. The authors give electronic temperatures for two different time delays in Fig. S5. This analysis needs to be done for all time delays to extract a carrier cooling time. Again, the analysis in Fig. S6 does not give this information. Niesner et al. (J. Am. Chem. Soc., 2016, 138 (48), pp 15717–15726) reported initial fast (300 fs) cooling, followed by slow cooling on a time scale of ~ 100 ps. Are the cooling dynamics (extracted from an analysis as in Fig. S5) consistent with the single-exponential temperature decay, which is assumed in the extraction of the cooling time?

As a consequence of mixing up the time scales, the authors identify the thermalization time as the critical time scale for a hot carrier solar cell in Fig. 4(c). From my understanding of a hot-carrier solar cell, this is wrong. The argument would apply if individual, high-energy carriers would be

harvested from the Boltzmann tail of the thermal distribution. The idea behind a hot-carrier solar cell, however, is that interacting carriers with the average thermal energy of the ensemble ($n \cdot kT$, where the prefactor n depends on details of the band structure and should be between 1.5 and 3) are harvested from a hot ensemble. Thus, the critical time scale for a hot-carrier solar cell is the one on which the ensemble cools down (and preserves a thermal energy above the one of the lattice) rather than the thermalization time (on which individual high-energy carriers maintain "optical" excess energies).

Also, it should be made clear for each individual data set shown, at which excitation density the data were recorded. E. g. in Fig. 3b the excitation density is not explicitly given. The distinction between the low and high carrier density is especially important in this study, since the initial carrier dynamics can be expected to be qualitatively different in the two situations. The statement "Interestingly, Hunsche et al. report no significant dependence of thermalization times on carrier density and excess energy for GaAs (18). The carrier thermalization times we observe for perovskites, however, show a strong dependence on both excess energy and carrier density" hence needs to be taken with some care. The authors compare two different excitation density regimes (below and above the "hot phonon bottleneck"), whereas the work by Hunsche et al. might well be done within one excitation regime. The current study does not give direct information on what the excitation density dependence of the energy relaxation rate would be within one regime, and especially not within the linear one below the hot phonon bottleneck or excitation densities of 10^{19} cm^{-3} , respectively.

2. My second concern is still on potential sample damage. In their response, the authors claimed no sample damage even at excitation densities as high as $2 \times 10^{19} / \text{cm}^3$. This is extraordinary as photo-induced damage has been reported by many authors at much lower excitation densities (see, e.g., Acc. Chem. Res. 2016, 49, 320–329; Nature Communications 2016, 7, 11574). The authors need to discuss why their samples are different.

3. The main conclusion of this manuscript is on ultrafast thermalization of hot carriers. The authors argued about low dielectric constant (I guess they meant optical dielectric constant) as compared to other inorganic semiconductors. However the dielectric constants of HOIPs are known to increase dramatically in the THz region (see, e.g., Mater. Horizons. 2016, 3, 1; J. Phys. Chem. C. 2015, 119, 5755). While the authors discussed related literature on phonon bottleneck, they should also address their findings in the context of recent reports (Science 2017, 356, 59; & Science 2016, 353, 1409) of long-lived hot carriers on time scales well beyond the carrier thermalization time presented here.

Reviewer #2 (Remarks to the Author):

Reviewing this revised version, I found that the authors have taken into account my comment and suggestions in a positive and constructive way to improve their ms. I therefore suggest to accept this version for publication in Nature Communications.

Reviewers' comments:

Reviewer #1 (Remarks to the Author):

While the revision addressed some of the questions, my major concerns remain.

We are happy that we were able to address some of the reviewer's questions, and we address the remaining helpful comments below.

1. My first concern is still on data analysis. The experimental approach was powerful, but the analysis and interpretation on several time scales and corresponding scattering processes got mixed up. Let me go into details.

(see attached drawing)

(1) Momentum scattering. This is what enters into the mobility in the Drude model. The Momentum scattering rate cannot be extracted from a 2D-TA experiment, since TA does not give momentum resolution. I agree with the authors, though, that the scattering rate observed in the experiment gives a lower limit for the momentum scattering rate. The difference between the two should nevertheless be made clear.

We agree with the reviewer, that 2D-TA cannot give momentum resolution. We also agree that our estimate for the mobility is a boundary indicating the impact of carrier-carrier scattering on charge transport. Following the reviewer's suggestion, we clarified this in the manuscript text and commented on the possibility of faster phonon scattering processes (page 10, line 1 – 3).

(2) Energy relaxation rate. This is what the authors measure in their analysis of the data by taking cuts at selected probe energies.

We agree with the reviewer that we are extracting energy relaxation times in our experiment.

(3) Thermalization time. It takes a large number of energy and momentum scattering events for the electrons to reach a thermal distribution.

We think that it only takes a few scattering events to reach a thermal distribution due to the special nature of our experiment, as we will explain in the following: In our experiment, we pump a broad energetic distribution of carriers at the same time. Carriers can therefore undergo carrier-carrier scattering with this large pool of carriers rather than only with carriers of the same excess energy (as would occur in the case of a pump-probe experiment with a spectrally narrow pump, which has been used previously to study carrier thermalization in semiconductors, Phys. Rev. B. 1993, 48, 17818–17826). It will therefore take only a few scattering events in order to reach a statistical distribution (which is here called a thermalized distribution). That this is true, is shown in figure 3c and d: The decay at $E_{\text{pump}} = E_{\text{probe}}$ is measuring energy relaxation (as explained by the reviewer), because we expect a carrier to leave the diagonal after one scattering event with a scattering partner of different energy and momentum. The rise of the signal at higher energies (above the diagonal of the 2D map) and at lower energies near the bandgap, however, is a measure for carrier thermalization. We observe that these two are happening at the same time indicating that a few scattering events are enough to form a thermal distribution.

The special character of our experiment is also representing the situation of a solar cell more accurately than a pump-probe experiment with a monochromatic pump beam due to the broadband nature of sunlight illumination.

Taking a look at Figure 3b, I would say that - if at all within the time window shown - a thermalized distribution is reached after 100 fs at the best.

We agree with the reviewer that reaching a thermal distribution is certainly an asymptotic process. It is therefore difficult to name one precise time to call a distribution thermalized. This was also a problem for works on thermalization in GaAs where people struggle to name this time precisely (e.g. Oudar et al. only comment on the spectral hole having completely disappeared after 4.2ps in Phys. Rev. Lett. 1985 55, 2074, while surely thermalization processes have happened before this time). We therefore decided to use the exponential decay time of the spectral hole as thermalization time. Our thermalization times therefore represent the half-time of the $E_{\text{pump}} = E_{\text{probe}}$ signal to decay and of the thermal distribution to rise (or more precisely $1/e = 0.37$, so the diagonal has decayed by 63%). This explains why in figure 3b, we observe a thermal distribution (fully statistical carrier distribution) after 50 - 100fs, but report a thermalization time (typical timescale of the process of carrier thermalization) of 40fs in figure 4b under these conditions.

The difference between the reviewer's comment and our draft is in the wording: **Carrier thermalization** is a process (in our manuscript clearly defined as the exchange of energy amongst carriers, page 5, line 22 – 23) whilst a **thermalized distribution** is the final state after completion of this process. So, the two pictures actually do not contradict each other. Whilst it is straight forward and objective to analyse the first (as we did via exponential rise times), it is rather challenging and can be highly subjective to define the latter. Referring to figure 3b, for example, one might call the spectrum at 48fs Fermi-like, whereas one could also argue that the distribution only reaches a fully thermal distribution at 100fs. This is further complicated by the presence of a bandgap renormalization process (Nature Communications 2015, 6, 8420) overlapping in this spectral region. We therefore decided that the 2DES **kinetics** are the most direct way to compare thermalization times for different excess energies and fluences. We added a discussion of this challenge to the manuscript (page 5, line 21 – 23).

We note that in our manuscript we are always clearly talking about the process of carrier thermalization and not about thermal distributions.

Attributing the curves in Fig. 3 c and d or the cross matrix elements in Fig. 4 to the thermalization time is speculative.

We don't follow the reviewer's criticism of the interpretation of figure 3c and d, in particular, since no clear argument for the arguably speculative nature of our attribution is given. For energies higher than the pump energy, we expect an early rise due to thermalization and a slower decay due to carrier cooling as seen in figure 3a. For energies near the bandgap, we expect an early rise due to carrier thermalization and a slower rise due to cooling as for example seen in the inset of figure 1b. Therefore, these kinetics show the process of carrier thermalization. (We note that cooling is not observed at the higher fluence in figure 3 c and d on the timescales of our experiment due to the hot phonon effect.)

For example, in figure 3c there is no further change in population after 50 fs which indicates that there is no further net exchange in energy and that thermalization is completed.

What one can see from those graphs is that the initial relaxation of the excess energy takes place on the time scale extracted. This does not give any information about whether the resulting carrier distribution is a thermalized one, which is a much more complicated question to answer. Thermalization times need to be extracted from plots like in Fig. 3b or S5.

We think that judging whether or not a spectrum is Fermi-like is very subjective and does not allow to accurately quantify the thermalization times. This is particularly important since we want to compare the changes in thermalization times with excess energy and fluence, and we believe that this relative comparison is independent of the measure used for thermalization time. The decay of the diagonal seemed therefore to be the most direct way to extract a thermalization timescale. We added a discussion of this challenge to the manuscript (page 5, line 21 – 23).

More importantly, cooling times need to be extracted from plots like in Fig. 3b, for example by analyzing the average energy or the high-energy tail of the distribution. Temperature (and average energy, which is often used as a measure of temperature) is an ensemble property, and the analysis in Fig. S6 does not give this information.

We agree that cooling times can only be extracted by analysing the high energy tail of the carrier distribution, as we have reported in our previous publication (Nature Communications 2015, 6, 8420). We did this in figure S5 where we did the temperature fitting for two time points. We decided to remove figure S6 from the supplementary information in order to avoid confusion. This figure was intended to only show the decay of hot carriers at one certain probe energy.

The aim of our manuscript is to study the very early carrier relaxation and not the carrier cooling which has been reported before. The time range accessed by our 2DES experiment does not span wide enough to measure the complete carrier cooling to room temperature which takes up to 100 ps.

(4) Cooling time. The authors give electronic temperatures for two different time delays in Fig. S5. This analysis needs to be done for all time delays to extract a carrier cooling time. Again, the analysis in Fig. S6 does not give this information. Niesner et al. (J. Am. Chem. Soc., 2016, 138 (48), pp 15717–15726) reported initial fast (300 fs) cooling, followed by slow cooling on a time scale of ~100 ps. Are the cooling dynamics (extracted from an analysis as in Fig. S5) consistent with the single-exponential temperature decay, which is assumed in the extraction of the cooling time?

We agree with the referee that cooling times require an ensemble analysis of the carrier energy distribution, as has been reported in multiple papers, also from our group. The time scales for cooling processes occur on much longer timescales than probed in the presented experiments, and do not require our 2D approach nor the 10-fs time resolution.

As a consequence of mixing up the time scales, the authors identify the thermalization time as the critical time scale for a hot carrier solar cell in Fig. 4(c). From my understanding of a hot-carrier solar cell, this is wrong. The argument would apply if individual, high-energy carriers would be harvested from the Boltzmann tail of the thermal distribution. The idea behind a hot-carrier solar cell, however, is that interacting carriers with the average thermal energy of the ensemble ($n \cdot kT$, where the prefactor n depends on details of the band structure and should be between 1.5 and 3) are harvested from a hot ensemble. Thus, the critical time scale for a hot-carrier solar cell is the one on which the ensemble cools down (and preserves a thermal energy above the one of the lattice) rather than the thermalization time (on which individual high-energy carriers maintain "optical" excess energies).

This is true for a semiconductor with weak electron-phonon coupling where ideally the cooling time is long compared to the carrier lifetime. In perovskites, however, the main cooling happens within the first 1ps compared to lifetimes of 100s of nanoseconds. Therefore, any background carrier population in the layer (e.g. due to non-perfect carrier extraction or due to photon recycling) will be in an equilibrium with the lattice. This cold population will then thermalize with any newly excited charge carriers and the ensemble temperature will barely rise. Therefore, the excess energy of carriers will be 'lost' due to its re-distribution.

This current confusion about perovskite hot extraction devices is the core of our argument.

Also, it should be made clear for each individual data set shown, at which excitation density the data were recorded. E. g. in Fig. 3b the excitation density is not explicitly given.

We agree with the reviewer that the fluences at which the experiments are performed are important parameters, and we point out that excitation densities were already explicitly given in all the figure captions of the manuscript.

The distinction between the low and high carrier density is especially important in this study, since the initial carrier dynamics can be expected to be qualitatively different in the two situations. The statement "Interestingly, Hunsche et al. report no significant dependence of thermalization times on carrier density and excess energy for GaAs (18). The carrier thermalization times we observe for perovskites, however,

show a strong dependence on both excess energy and carrier density" hence needs to be taken with some care. The authors compare two different excitation density regimes (below and above the "hot phonon bottleneck"), whereas the work by Hunsche et al. might well be done within one excitation regime. The current study does not give direct information on what the excitation density dependence of the energy relaxation rate would be within one regime, and especially not within the linear one below the hot phonon bottleneck or excitation densities of 10^{19} cm^{-3} , respectively.

We agree with the reviewer that the hot phonon effect could also affect carrier thermalization if carrier-phonon scattering contributes to the carrier thermalization. We now comment on this in the manuscript (page 9, line 5 – 9). The comparison with Hunsche et al. is still valid when considering the excess energy dependence which is very strong in our data (figure 4b) but apparently not present in Hunsche's experiments.

2. My second concern is still on potential sample damage. In their response, the authors claimed no sample damage even at excitation densities as high as $2 \times 10^{19} / \text{cm}^3$. This is extraordinary as photo-induced damage has been reported by many authors at much lower excitation densities (see, e.g., Acc. Chem. Res. 2016, 49, 320–329; Nature Communications 2016, 7, 11574). The authors need to discuss why their samples are different.

We have already provided experimental evidence for sample stability in our resubmission, as well as detailed information on the measurement conditions and sample fabrication in our initial submission. This information should be helpful to researchers in the future to achieve measurement conditions under which reliable experiments can be performed.

In the cited publications, the sample damage was observed under continuous wave (cw) illumination (e.g. photo-current degradation over 120min, Nature Communications 2016, 7, 11574; voltage rise and current decrease over 30min, Acc. Chem. Res. 2016, 49, 320–329). Our experiment is, however, performed under completely different conditions, i.e. pulsed excitation with a low repetition rate of 1kHz. The average pump power put into the system is therefore similar in our case (power density of $\sim 30 \text{ mW/cm}^2$ at highest fluence) to cw illumination with one sun ($\sim 75 \text{ mW/cm}^2$) in the cited literature. Additionally, many of the reported light-activated effects are reversible effects, likely due to ion migration and trap formation (Nature Communications 2016, 7, 11574; Acc. Chem. Res., 2016, 49 (2), 286–293). In our pulsed experiment, charges will only have a lifetime of 100s of picoseconds at the investigated fluences (Nature Communications 2016, 7, 13941). Due to the low repetition rate, the film will therefore have ~ 1 millisecond to recover after each pulse.

We would also like to point towards other literature on ultrafast spectroscopy on perovskites which was performed under similar excitation densities (e.g. $1.5 \times 10^{19} \text{ cm}^{-3}$ in Nature Communications 2017, 8, 14350; $\sim 1 \times 10^{20} \text{ cm}^{-3}$ in J. Phys. Chem. Lett., 2015, 6 (1), 153–158; $6 \times 10^{18} \text{ cm}^{-3}$ in Nature Communications 2015, 6, 8420 and Nature Photonics 2016, 10, 53 - 59).

3. The main conclusion of this manuscript is on ultrafast thermalization of hot carriers. The authors argued about low dielectric constant (I guess they meant optical dielectric constant) as compared to other inorganic semiconductors. However the dielectric constants of HOIPs are known to increase dramatically in the THz region (see, e.g., Mater. Horizons. 2016, 3, 1; J. Phys. Chem. C. 2015, 119, 5755).

We agree with the reviewer that the optical dielectric constant is the relevant parameter. We changed the wording in the manuscript from 'high-frequency' to 'optical dielectric constant' to be more precise (page 8, line 19). With respect to the cited literature, Walsh reports a theoretical optical dielectric constant of $\epsilon = 5$ (which is even lower than the value we used) and a value of $\epsilon = 25$ at 1THz. Considering the timescales which we are probing, only screening with a frequency of >100 THz plays a role. The cited value of $\epsilon = 8$ taken at 400 THz seems therefore like a good estimate to us (Nature Photonics 2015, 9, 106 – 112).

While the authors discussed related literature on phonon bottleneck, they should also address their findings in the context of recent reports (Science 2017, 356, 59; & Science 2016, 353, 1409) of long-lived hot carriers on time scales well beyond the carrier thermalization time presented here.

We thank the reviewer for recommending this literature to us. We added a discussion of this literature to the discussion of hot carrier solar cells (page 9, line 7 – 19), whilst we think that they are less relevant for the process of carrier thermalization.

Reviewer #2 (Remarks to the Author):

Reviewing this revised version, I found that the authors have taken into account my comment and suggestions in a positive and constructive way to improve their ms. I therefore suggest to accept this version for publication in Nature Communications.

We thank the reviewer for this positive assessment of our revised manuscript, and we are happy that we were able to include the reviewer's helpful comments and suggestions in a positive and constructive way.

Reviewers' comments:

Reviewer #1 (Remarks to the Author):

The revision has addressed most of my concerns. I recommend publication.